# The Carbon-Isotope Record of the Sub-Seafloor Biosphere

**Patrick Meister [1,]\* and Carolina Reyes [2]** 

[1] Department of Geodynamics and Sedimentology, University of Vienna, Althanstr. 14, 1090 Vienna, Austria
[2] Department of Environmental Geosciences, University of Vienna, Althanstr. 14, 1090 Vienna, Austria; creyes6@gmail.com
\* Correspondence: patrick.meister@univie.ac.at

**Abstract:** Sub-seafloor microbial environments exhibit large carbon-isotope fractionation effects as a result of microbial enzymatic reactions. Isotopically light, dissolved inorganic carbon (DIC) derived from organic carbon is commonly released into the interstitial water due to microbial dissimilatory processes prevailing in the sub-surface biosphere. Much stronger carbon-isotope fractionation occurs, however, during methanogenesis, whereby methane is depleted in $^{13}C$ and, by mass balance, DIC is enriched in $^{13}C$, such that isotopic distributions are predominantly influenced by microbial metabolisms involving methane. Methane metabolisms are essentially mediated through a single enzymatic pathway in both *Archaea* and *Bacteria*, the Wood–Ljungdahl (WL) pathway, but it remains unclear where in the pathway carbon-isotope fractionation occurs. While it is generally assumed that fractionation arises from kinetic effects of enzymatic reactions, it has recently been suggested that partial carbon-isotope equilibration occurs within the pathway of anaerobic methane oxidation. Equilibrium fractionation might also occur during methanogenesis, as the isotopic difference between DIC and methane is commonly on the order of 75‰, which is near the thermodynamic equilibrium. The isotopic signature in DIC and methane highly varies in marine porewaters, reflecting the distribution of different microbial metabolisms contributing to DIC. If carbon isotopes are preserved in diagenetic carbonates, they may provide a powerful biosignature for the conditions in the deep biosphere, specifically in proximity to the sulphate–methane transition zone. Large variations in isotopic signatures in diagenetic archives have been found that document dramatic changes in sub-seafloor biosphere activity over geological time scales. We present a brief overview on carbon isotopes, including microbial fractionation mechanisms, transport effects, preservation in diagenetic carbonate archives, and their implications for the past sub-seafloor biosphere and its role in the global carbon cycle. We discuss open questions and future potentials of carbon isotopes as archives to trace the deep biosphere through time.

**Keywords:** carbon isotopes; deep biosphere; diagenetic carbonates; methanogenesis; anaerobic methane oxidation; Wood–Ljungdahl pathway

---

## 1. Introduction

Carbon, in its reduced form, is not only the essential building material of life, but due to its large isotope variations, it can also serve as a tracer of biogeochemical processes in the environment and as an indicator of the state of the global carbon cycle. During assimilation of $CO_2$ to organic matter in the water column, carbon is depleted in $^{13}C$ by 20–30‰; however, variations in $\delta^{13}C$ in ocean and atmosphere are usually in the few-permil range. These variations are essentially balanced by input to and output from the ocean and atmosphere, and they only change upon variations in rates of primary production and burial of organic carbon relative to inorganic carbon (Broeker, 1970) [1].

In contrast, isotopic compositions of dissolved carbon species in marine sedimentary porewater show a large range of values. These systems, which are part of a deep biosphere, are exclusively inhabited by Prokaryotes, whereby the "deep subsurface biosphere" has been operationally defined as an "ecosystem that persists at least one metre, if not more" (Edwards et al., 2012) [2]. The deep biosphere is organized as zones of different metabolic activity in the sequence of downward decreasing redox potential (Froelich et al., 1979) [3], whereby the presence and extension of these zones vary considerably in different regions of the ocean (D'Hondt et al., 2004) [4]. Sulphate reduction represents the most abundant anaerobic respiration process, and, although dissimilatory degradation of organic matter generally exhibits small fractionation effects (e.g., Hayes et al., 1989) [5], it delivers dissolved inorganic carbon (DIC) strongly depleted in $^{13}C$ to the porewater. Furthermore, large fractionation effects are observed during fermentation reactions where the pools of $CO_2$ and $CH_4$ are involved. Biogenic methane often shows $\delta^{13}C$ values as negative as −100‰ relative to the Vienna Peedee Belemnite (VPDB) standard (Claypool and Kaplan, 1974) [6], while positive values of up to +35‰ have been observed in dissolved inorganic carbon (DIC) related to methane production (Heuer et al., 2009) [7]. In turn, strongly $^{13}C$-depleted DIC is produced as a result of anaerobic oxidation of methane (AOM). Ultimately, the different metabolic processes result in a large variability in inorganic carbon, especially in proximity to the sulphate–methane transition zone.

Inorganic carbon can be incorporated in diagenetic carbonates and thereby become preserved as part of the mineral phase. Often, diagenetic carbonates form as a result of microbial dissimilatory activity, which also contributes to global carbon burial (Schrag et al., 2013) [8]. Diagenetic carbonates can trap the isotopic signature of the surrounding fluid from which they were precipitated, and preserve it for millions of years as part of the geological record. As opposed to carbon isotopes in marine carbonates, which vary in $\delta^{13}C$ in the few-permil range and can be used to reconstruct the global carbon cycle, diagenetic carbonates can show large variations based on type and rate of microbial activity and provide information on past biogeochemical conditions at a specific location. Indeed, previous studies have shown that carbon-isotope values preserved in diagenetic carbonates can be used as an archive of past deep biosphere activity (e.g., Kelts and McKenzie, 1984; Malone et al., 2002; Meister et al., 2007; Meister, 2015) [9–12]. Recent studies also showed that other light isotope systems (i.e., isotopes of light elements) in diagenetic phases, e.g., sulphur isotopes preserved in diagenetic pyrite (Meister et al., 2019a) [13], provide evidence similar to carbon isotopes, indicating that in the past, the conditions in the deep biosphere were different from today, and that biogeochemical zones migrated upwards and downwards in the sediment. Despite these insights, currently there are several problems that hamper a more detailed interpretation of diagenetic carbon-isotope records: (1) Fractionation effects are incompletely understood; (2) diffusive mixing and non-steady state conditions result in a complex mixture of isotopic compositions from different sources; and (3) models to quantitatively predict carbonate precipitation are not sufficiently developed.

Here, we provide a brief overview of the current state of knowledge of carbon-isotope effects related to the main anaerobic metabolic processes, sulphate reduction, methanogenesis, and AOM, occurring in marine sediments (Section 2). We discuss the different fractionation mechanisms, as part of enzymatic pathways, including a distinction between kinetic and potential equilibrium fractionation effects. In Section 3, we assess how different microbial processes affect carbon-isotope profiles in DIC and $CH_4$ in marine porewaters and how these profiles are subject to diffusive mixing and advective transport. Section 4 is focused on the controls on diagenetic carbonate formation in the sub-seafloor biosphere and how combined effects of carbon-isotope fractionation, transport, and mineral precipitation result in diagenetic carbon-isotope records. In Section 5, we discuss the importance of diagenetic carbonate burial on secular variation in the global carbon cycle and its $^{13}C$ signature and how we can trace these variations back in time to reconstruct the evolution of a dynamic sub-seafloor biosphere through Earth's history.

## 2. Carbon-Isotope Fractionation by Different Microbial Pathways

### 2.1. Kinetic Fractionation

Stable isotope fractionation strongly depends on the molecular pathways of metabolic reactions and is mostly the result of a kinetic effect, whereby a molecule having the same thermal vibration energy but higher mass is less likely to overcome the activation energy barrier of the reaction (Hoefs, 2018) [14]. Molecular pathways are mostly known in detail, but it remains unclear at which step of the pathway the isotope fraction occurs. Hence, from an isotopic point of view, many metabolic reactions represent a black box, and fractionation effects have been determined for the overall reactions.

In practice, two approaches have been used to determine microbial carbon-isotope fractionation effects: First, measuring carbon-isotope distributions in metabolites, such as $CH_4$, DIC, and other organic intermediates dissolved in natural porewaters can provide an "apparent" fractionation factor. Second, carbon-isotope fractionation factors have been determined directly from microcosm (Alperin et al., 1992) [15] or pure culture experiments. By convention, the fractionation factor $\alpha$ is defined as the isotope ratio of the reactant divided by the isotope ratio of the product. For convenience, fractionation can also be expressed by the separation factor, which is the difference in permil of the $\delta^{13}C$ of reactant and the product ($\varepsilon = (\alpha - 1) \cdot 1000$; Hayes, 1993; 2004) [16,17].

*Sulphate reduction*: Fractionation during dissimilatory sulphate reduction (Equation (1)) is considered insignificant (e.g., Claypool and Kaplan, 1974) [6].

$$2\,[CH_2O] + SO_4^{2-} \rightarrow HS^- + 2\,HCO_3^- + H^+ \tag{1}$$

Assuming that average organic matter $[CH_2O]$ is consumed quantitatively, while intermediate pools are very small, essentially no fractionation effects should be expected in natural sediments. In addition, fractionation factors determined in culture experiments using defined substrates for dissimilatory sulphate reduction (Londry and Des Marais, 2003) [18] were near to one.

*Anaerobic methane oxidation*: Whiticar and Faber (1986) and Alperin et al. (1988) [19,20] showed that a small kinetic fractionation ($\alpha = 1.0088$) occurs if sulphate reduction is linked to the anaerobic oxidation of methane (AOM; Equation (2)), resulting in a $^{13}C$ enrichment of the residual $CH_4$ pool.

$$CH_4 + SO_4^{2-} \rightarrow HS^- + HCO_3^- + H_2O \tag{2}$$

AOM culture experiments yielded significantly higher kinetic fractionation factors ($\alpha = 1.012–1.039$) than were evident from field observations (e.g., Holler et al., 2009) [21]. The reason for this discrepancy is unclear at present.

*Methanogenesis*: Fractionation effects during methanogenesis vary over a large range dependent on the enzymatic pathway and experimental conditions as shown in the histogram of separation factors compiled from the literature (Figure 1). During acetoclastic methanogenesis:

$$2\,CH_3COOH \rightarrow CO_2 + CH_4 \tag{3}$$

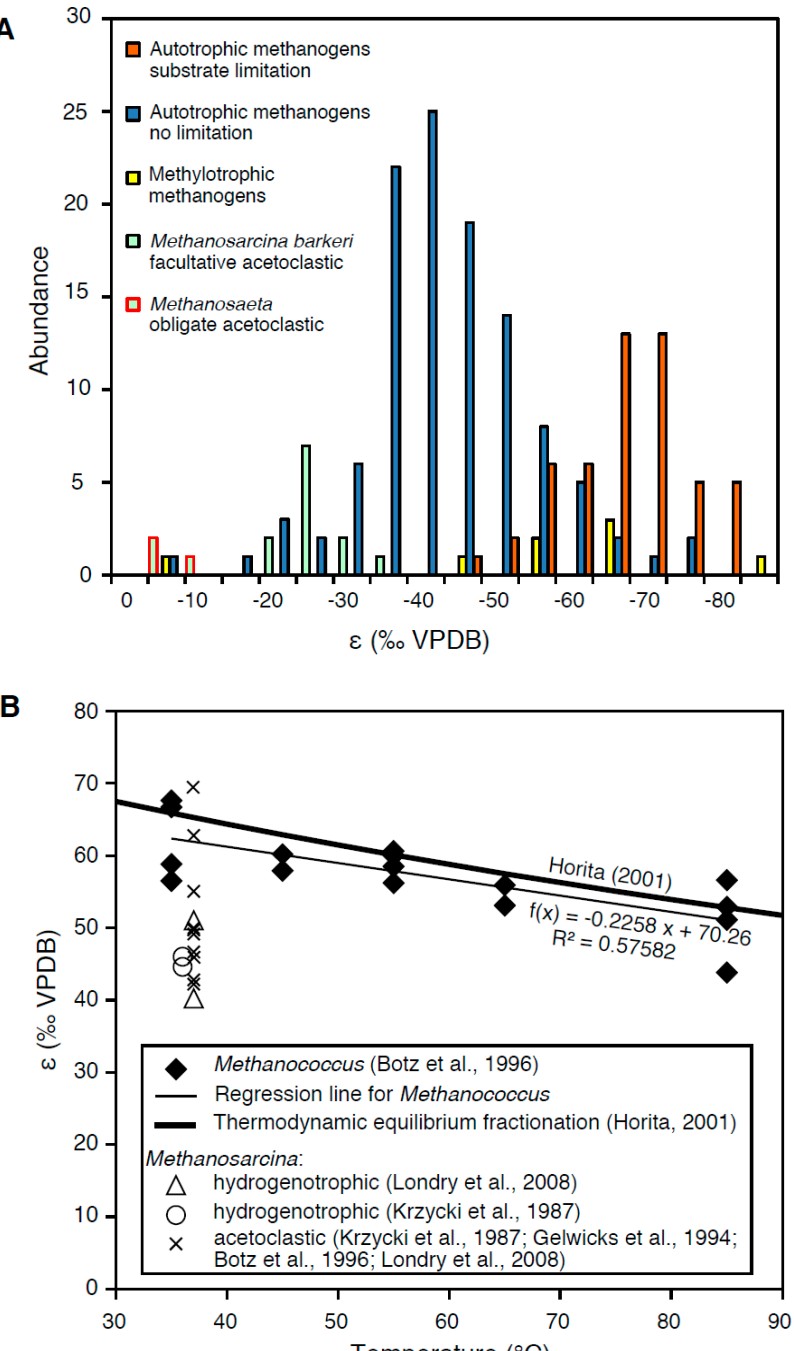

**Figure 1.** Compilation of separation factors *ε* from culture experiments using species of *Methanosarcina*, *Methanococcus*, *Methanobacterium*, *Methanothermobacter*, and *Methanothermococcus* grown under different conditions: (**A**) Histogram showing the abundance distribution of separation factors for methanogens grown with acetate, hydrogen, and methanol/trimethylamine. Distinctively larger isotope effects are observed during hydrogenotrophic growth, in particular, under substrate limitation, whereas minor fractionation occurs during acetoclastic growth. Thereby, obligate acetoclasts show insignificant fractionation. Data are compiled from Games et al. (1978) [22]; Fuchs et al. (1979) [23]; Belyaev et al. (1983) [24]; Balabane et al. (1987) [25]; Krzycki et al. (1987) [26]; Gelwicks et al. (1994) [27]; Botz et al. (1996) [28]; Summons et al., 1998 [29]; Valentine et al. (2004) [30]; Londry et al. (2008) [31]; Okumura et al., (2016) [32]; and Miller et al. (2018) [33]. (**B**) Temperature dependence of separation factors in experiments with *Methanococcus* growing under $H_2$ limitation (Botz et al., 1996) [28] in comparison to the thermodynamic isotope equilibrium from Horita (2001) [34].

which is a fermentation reaction, the observed fractionation factor between reactant (acetate) and product ($CH_4$) is small. While fractionation of the carboxyl carbon may be uncoupled from the methyl carbon, Sugimoto and Wada (1993) [35] showed in sediment incubation experiments that this reaction only produces a minor intramolecular isotopic difference. Pure-culture incubations with the facultative acetoclast *Methanosarcina barkeri* by Krzycki et al. (1987) [26], Gelwicks et al. (1994) [27], and Londry et al. (2008) [31] show a fractionation effect in the range of −20‰ to −30‰, in contrast to the obligate acetoclast *Methanosaeta thermophila*, showing insignificant fractionation (Valentine et al., 2004 [30]; Figure 1A).

Much larger fractionation effects were found for hydrogenotrophic methanogenesis (Whiticar et al., 1986) [36] with an apparent fractionation factor $\alpha$ = 1.05–1.09. However, it needs to be taken into account that for each mole of $H_2$ produced by fermentation of organic matter also 0.5 mol of $CO_2$ are produced, following the overall reaction:

$$2\,[CH_2O] \overset{+2H_2O}{\rightarrow} 2\,CO_2 + 4\,H_2 \overset{-2H_2O}{\rightarrow} CO_2 + CH_4 \tag{4}$$

Therefore, the apparent separation factors determined in the field do not provide the exclusive fractionation factors of the pure methanogenic step but include the $CO_2$ from fermentation. Methanogenic culture experiments have been performed with *Methanosarcina barkeri* (Games et al., 1978 [22]; Krzycki et al., 1987 [26]; Londry et al., 2008 [31]), an acetoclastic methanogen capable of growing autotrophically, as well as with several strains of the obligate autotrophic methanogens *Methanococcus* (Botz et al., 1996) [28], *Methanobacterium* (Games et al., 1978 [22]; Fuchs et al., 1979 [23]; Belyaev et al., 1983 [24]; Balabane et al., 1987 [25]; Okumura et al., 2016 [32]; Miller et al., 2018 [33]), *Methanothermobacter* (Valentine et al., 2004 [30]; Okumura et al., 2016) [32], and *Methanothermococcus* (Okumura et al., 2016) [32] under conditions supplied with $H_2$ and $CO_2$. As shown by the histogram in Figure 1A, distinctly larger separation factors ($\varepsilon$) are observed for hydrogenotrophic than for acetoclastic methanogenesis, independent of the organism used. Also, a great range of separation factors is observed during methylotrophic growth, using *M. barkeri* or *Methanococcoides burtonii* supplied with methanol or trimethylamine (Summons et al., 1998 [29]; Londry et al., 2008 [31]; data included in Figure 1A).

Using a flow-through fermentor system, Botz et al. (1996) [28] were able to maintain growth of different species of *Methanococcus* under substrate limitation, which generally resulted in larger separation factors. A similar effect was achieved using a fermenting coculture, providing limited hydrogen substrate (Okumura et al., 2016 [32]), or under strongly alkaline conditions (Miller et al., 2018 [33]). Orange bars in Figure 1A highlight the significantly larger isotope effects under substrate-limited conditions. These conditions may better resemble the natural system, where often substrate limitation prevails and organisms grow extremely slowly (cf. "the starving majority"; Jørgensen and D'Hondt, 2006) [37].

## 2.2. Equilibrium Fractionation

Not entirely clear is the observation that methane often shows extremely negative $\delta^{13}C$ values at sulphate–methane transition zones (SMTZ), i.e., at the depth at which it is almost entirely consumed. If this were the result of kinetic fractionation, $CH_4$ would be expected to show less negative values (because light $CH_4$ is preferentially oxidized). Even though autotrophic methanogenesis may contribute to lower values, the isotope effect of AOM should dominate.

Recently, Holler et al. (2009, 2011) [21,38] and Yoshinaga et al. (2014) [39] published results of experimental studies using mixed cultures of sulphate-reducing bacteria and methanotrophic archaea, forming a consortium capable of anaerobically oxidizing methane. They suggest that the AOM reaction is reversible, as an amount of dissolved inorganic carbon (DIC) is channelled back into the methane pool. This was demonstrated by addition of $^{14}C$-labelled DIC, which ended up in the methane during the experiment. This finding is remarkable since it provides insight into the enzymatic

pathway. Yoshinaga et al. (2014) [39] suggested that a partial isotopic equilibration occurs within this process, and this isotopic fractionation could explain the formation of methane with strongly negative $\delta^{13}C_{CH4}$ values at the sulphate–methane transition (SMT) zones in marine sediments. In the study of Yoshinaga et al. (2014) [39], a kinetic fractionation model was used, where kinetic fractionation occurs in both forward and backward directions and a steady state is reached in a diffusion-limited system (cf. Hayes et al., 2004 [17]). Since isotopic equilibrium can only be reached if true chemical equilibrium occurs (Urey and Greiff, 1935) [40], it is assumed that also the AOM reaction under natural porewater conditions is close to thermodynamic equilibrium. This is indeed the case as the free energy yield of AOM is minimal and depends on the concentration levels of reactant and product. While abiotically, $CO_2$ does not exchange isotopes with $CH_4$ under Earth surface temperature (Giggenbach, 1982) [41], the equilibrium fractionation effect between $CH_4$ and $CO_2$ is extrapolated from high temperature (Richet et al., 1977 [42]; Horita, 2001 [34]). The separation factor is rather large, on the order of 70‰ at ambient temperature, and indeed, such a large isotopic difference is observed in marine porewaters.

The idea of equilibrium carbon-isotope fractionation is not entirely new. It was already suggested by Bottinga (1969) [43] for methanogenesis and is supported by culture experiments (Valentine et al., 2004 [30]; Penning et al., 2005 [44]; Moran et al., 2005 [45]; Takai et al., 2008 [46]). While experiments performed under different conditions show large scatter in the separation factors, the experiments under substrate limitation show significantly larger separation factors (Figure 1A). Plotting exclusively the values from experiments with *Methanococcus* by Botz et al. (1996) [28] against incubation temperature (Figure 1B), whereby the initial time steps were omitted, results in a regression line that fits very well to the thermodynamic equilibrium separation factor from Horita (2001) [34]. These experiments strongly suggest that also during methanogenesis, isotopic equilibration occurs.

## 2.3. Potential Fractionation Effects within the Molecular Pathway

While radiotracer experiments provide valuable information on fractionation mechanisms, these findings can now be considered to discuss fractionation effects within the actual biochemical pathways. We focus here on the Wood–Ljungdahl pathway (WL pathway; also known as acetyl coenzyme-A pathway), which is the essential pathway used by methanogens, acetogens, as well as methanotrophs. The WL pathway is at the centre of all methane metabolisms and, therefore, should be mainly responsible for carbon-isotope fractionation in natural aquatic sediments.

*The Wood-Ljungdahl pathway*: The backbone of the WL pathway is shown in Figure 2. It starts with $CO_2$ being first reduced to formate by formate-dehydrogenase (FDH). Alternatively, methylated substrates, such as methanol or methylamine, may be used. In *Archaea*, the formate is further reduced along a cascade of reduction steps to methyl-tetrahydromethanopterin (4H-MTP). In *Bacteria*, tetrahydrofolate is used instead of 4H-MTP. At the core of the WL pathway, acetyl-coenzyme-A synthase (ACS) combines a methyl group, containing a reduced carbon, with a carboxyl carbon when forming acetyl-coenzyme-A. The carboxyl carbon is supplied from $CO_2$ via a carbon monoxide-dehydrogenase (CODH), which is part of the ACS-CODH cluster (also called CO-methylating acetyl-CoA synthase; Adam et al. 2018 [47]). The pathway ends with the production of acetate or may continue from acetyl-CoA further to biosynthesis. In methanogens, the methyl group from methyl-4H-MTP is passed to coenzyme-M (CoM) and released as $CH_4$.

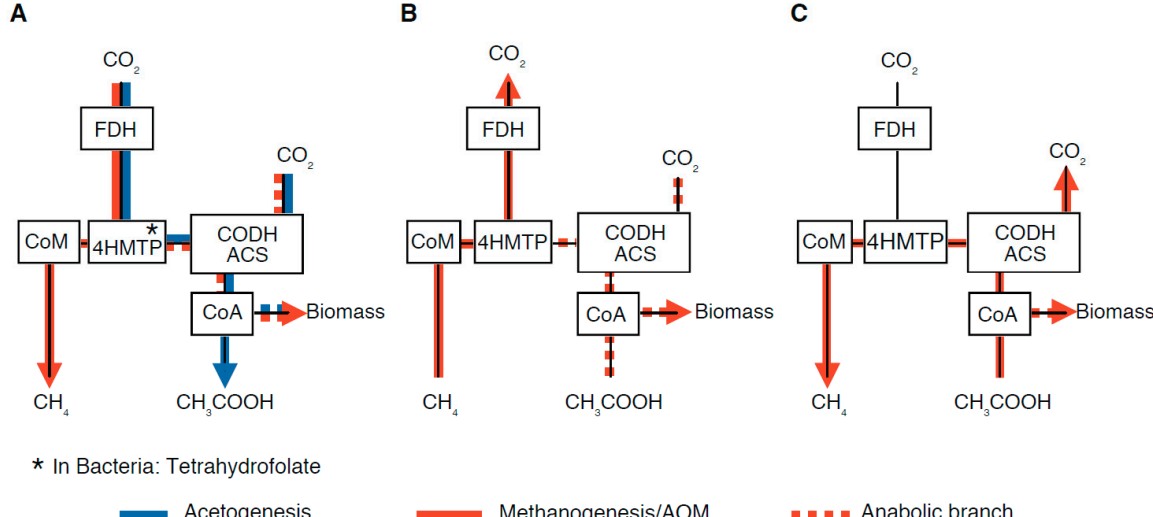

**Figure 2.** Wood–Ljungdahl pathway (black lines) and different directions in which metabolic reactions can run through the pathway: (**A**) Autotrophic methanogenesis (solid red line) and acetogenesis (blue line; via Tetrahydrofolate); (**B**) anaerobic methane oxidation; (**C**) acetoclastic methanogenesis. Also, the biosynthetic branch is indicated by dashed red lines. The scheme is drawn according to the WL pathway shown in Martin and Russel (2007) [48]. Abbreviations: Acetyl-CoA synthase: ACS; carbon monoxyde dehydrogenase: CODH; coenzyme-A: CoA; coenzyme-M: CoM; formate-dehydrogenase: FDH; tetrahydro-methanopterin: 4HMTP.

While the WL-pathway represents the most ancient pathway of carbon fixation that still exists in modern *Archaea* and *Bacteria* (cf. Weiss et al., 2016) [49], different organisms run the pathway in different directions, depending on the biogeochemical conditions, to gain energy and biomass. Figure 2 shows the three directions in which different organisms use the pathway. Autotrophic (i.e., hydrogenotrophic) methanogens such as *Methanococcus* (Figure 2A) use the MTP branch to reduce carbon and then pass the methyl group to CoM. However, they also use the ACS-CODH cluster for anabolism (dashed red line; Berghuis et al., 2019) [50]. Acetogens follow the same route (blue lines), but they do not produce CoM.

In the second case (Figure 2B), methanotrophs use CoM and the MTP-chain to oxidize $CH_4$ to $CO_2$ by running the pathway backwards. Nevertheless, for biosynthesis they still rely on carboxyl-carbon, which is directly fixed via the ACS-CODH cluster and, hence, not derived from methane. This has been confirmed by radiotracer experiments (Kellermann et al., 2012) [51], although isotopically light DIC that is delivered by AOM may be directly used within the community cluster (Alperin et al., 2009) [52]. Besides, the same pathway is also used by methanotrophic bacteria (Skennerton et al., 2017) [53].

Methanogens, using acetate as a substrate (acetoclastic methanogens, such as *Methanosarcina*), run the WL-pathway sideways (Figure 2C; see Thauer et al., 2008) [54]. They use the ACS-CODH cluster in reverse to cleave the carboxyl group from acetate and oxidize it to $CO_2$, while the methyl group is transferred to 4H-MTP and CoM.

*Isotope fractionation in the WL-pathway*: Having established the general pathways, we can now consider, where in the pathway the essential fractionation effect occurs. Valentine et al. (2004) [30] discuss the concept of differential reversibility within the WL pathway, where the hydrogenotrophic methanogenic steps from $CO_2$ to 4H-MTP are reversible but the methyl transfer to CoM represents a kinetic bottleneck (Figure 3). This step would only become reversible under strong substrate limitation, and indeed, isotopic equilibration seems to occur under such conditions as evidenced by separation factors of substrate-limited autotrophic (hydrogenotrophic) methanogens in Figure 1A,B. It is also conceivable that in the opposite direction, during AOM, the methyl transfer from CoM to 4H-MTP (Figure 3) would be the critical step, becoming reversible under electron acceptor limitation.

Meister et al. (2019b) [55] propose a model, where isotopic equilibration in fact occurs within the pathway, which would most likely be localized at this particular step.

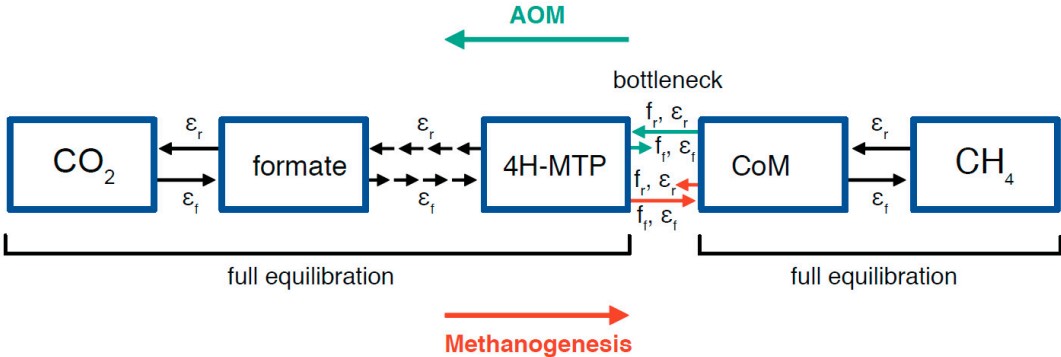

**Figure 3.** Scheme showing the differential reversibility of steps in the methanopterin-branch of the autotrophic methanogenic Wood–Ljungdahl pathway, as proposed by Valentine et al. (2004) [30]. The transfer of methyl groups by methyl-transferase to coenzyme-M may represent the bottleneck in the entire pathway, only becoming reversible under strong substrate limitation.

While differential reversibility could explain isotope fractionation for hydrogenotrophic methanogens, fractionation during acetoclastic methanogenesis (Figure 1A) is very small and not anywhere near isotopic equilibrium. Isotopic fractionation during acetoclastic methanogenesis could also not explain the large difference in $\delta^{13}C$ between $CH_4$ and $CO_2$ in marine sediments, while the intra-molecular difference between the methyl and carboxyl carbons in acetate is only in the range of 7‰–14‰ (Blair and Carter, 1992; Sugimoto and Wada, 1993) [35,56]. While acetate is split into a methyl-group and a carboxyl-group within the ACS-CODH cluster, a direct exchange of the methyl-C and the carboxyl-C would not be possible, due to the high energy barrier involved in cleaving the C=O double bond. Also, in most cases, acetate represents a small intermediate pool and cannot significantly fractionate against $CH_4$ and $CO_2$. Fractionation in the anabolic branch (Figure 2) is also not likely to be a major cause of $^{13}C$-depleted methane, since archaeal lipids are themselves depleted in $^{13}C$ (e.g., Hinrichs et al., 2000; Pancost et al., 2000; Contreras et al., 2013) [57–59]. Isotope exchange may still occur through the MTP branch in facultative hydrogenotrophic *Methanosarcina* (capable of hydrogenotrophic and acetoclastic methanogensis). Indeed, *Methanosarcina* growing acetoclastically shows 20‰–30‰ fractionation, while fractionation in the obligate acetoclastic *Methanosaeta*, which is a common organism in marine sediments (Carr et al., 2017) [60], is entirely insignificant (Figure 1A).

Overall, isotopic equilibration through the WL pathway may represent the most predominant carbon-isotope fractionation mechanism in natural marine sediments. However, this concept requires further investigation. Radiotracer experiments with methanogens or incubations with purified enzymes could clarify some of the fundamental fractionation mechanisms.

### 2.4. The Effects of Substrate and Carbon Limitation

*Energy limitation*: It is well known that microorganisms in the deep biosphere often operate under extreme energy limitation (Jørgensen and D'Hondt, 2006) [37]. Under such conditions, enzymatic reactions are likely reversible, leading to isotopic exchange. An isotopic equilibration is possible as long as the reaction is also approaching a chemical equilibrium (Urey and Greiff, 1935) [40]. This is clearly the case for hydrogenotrophic methanogens under $H_2$ limitation, while ample inorganic carbon may still be available. This is also the case during AOM because the energy yield of AOM is minimal and subject to concentration levels of methane and sulphate. Yoshinaga et al. (2014) [39] observed an increasing reverse AOM flux and accordingly, more complete isotopic equilibration with decreasing sulphate concentration. In their model, equilibrium fractionation results from the difference in fractionation of the forward and reverse reactions within a diffusion-limited system.

*Organic carbon substrate limitation*: Also, organic carbon substrates are often limiting under deep-biosphere conditions. In particular, small molecules, such as acetate or methyl groups, are common intermediates produced by other, fermenting organisms at rather low rates. Under sluggish turnover rates, not only is the energy source limited, but also the carbon source, in which case, complete turnover may result in a minimal fractionation effect.

*Inorganic carbon limitation*: It is very unlikely that in normal marine sediments dissolved inorganic carbon (DIC) becomes limiting. This is because, according to Equation (4), for each mole of $H_2$ produced by fermentation of organic matter, approximately half a mole of $CO_2$ is produced (i.e., two moles of $CO_2$ per mole of $CH_4$). $CO_2$-limitation may, however, occur if an excess of $H_2$ is supplied from abiotic sources. Such observations have been made in off-ridge hydrothermal systems, such as the Lost-City hydrothermal field in the Atlantic, where large amounts of $H_2$ are produced due to alteration of ultramafic rocks by seawater (serpentinization; see McCollom and Bach, 2009) [61]. Ultramafic fluids are often highly alkaline and conducive to intense carbonate precipitation if they come in contact with seawater, i.e. their DIC content is usually very low. Under high $H_2$ content and concomitantly low DIC content, the system may indeed become DIC-limited. Under $CO_2$-limitation a complete Rayleigh effect would occur and, hence, isotope fractionation would become small. In these cases, biogenic methane production would produce $CH_4$ with a rather high $^{13}C$ content, thus mimicking an isotopic signature of methane that was generally thought to be of abiotic origin (Meister et al., 2018) [62]. The experiments by Miller et al. (2018) [33] seem to confirm that methanogenesis under alkaline conditions with strong Ca-carbonate supersaturation indeed produce methane with very small fractionation relative to $CO_2$.

Considering all these different effects, we may conclude that carbon-isotope equilibrium fractionation is most likely to prevail in normal marine sediments.

## 3. Carbon-Isotope Distribution in Porewater Systems

### 3.1. Isotope Effects in Diffusive Mixing Profiles

While the exact mechanisms of fractionation remain incompletely understood, and further experiments are necessary to shed light on this matter, a further aspect that needs consideration is how the fractionation manifests in isotopic differences in natural methane and DIC profiles in sedimentary porewater. In unlithified marine sediments, porosity is usually large, allowing for diffusive transport of solutes along concentration gradients. While advective transport may substantially contribute to methane and DIC transport (see Section 3.4), diffusive mixing alone leads to considerable complexity in isotopic distributions. The main C-constituents in sedimentary porewaters are DIC and methane. Although dissolved organic carbon (DOC) compounds can be detected, which are largely fermentation products, such as acetate, serving as intermediates in microbial metabolic networks, their concentration gradients are generally too small to give rise to significant diffusive transport. Even though their turnover may exhibit substantial fractionation effects (e.g., Heuer et al., 2008 [7]; Ijiri et al., 2012 [63]), these effects do not manifest in the end products if these compounds undergo quantitative production and consumption, resulting in a complete Rayleigh effect.

*Diffusive mixing between seawater and porewater*: Inorganic carbon dissolved in seawater has a $\delta^{13}C$ around 0‰, but its concentration is only on the order of 2 mmol/L. DIC concentration often steeply increases with depth in the sediment, often reaching 10 mmol/L within a few centimetres. Most DIC in the sediment is derived from dissimilatory degradation of organic carbon, exhibiting an isotopic composition on the order of −20‰, while the influence of the small amounts of DIC from seawater rapidly decreases with depth (Figure 4A). Diffusive mixing is then manifested as a mixing hyperbola, resulting in rather negative values at shallow depth in a diffusive boundary layer (Zeebe, 2007) [64].

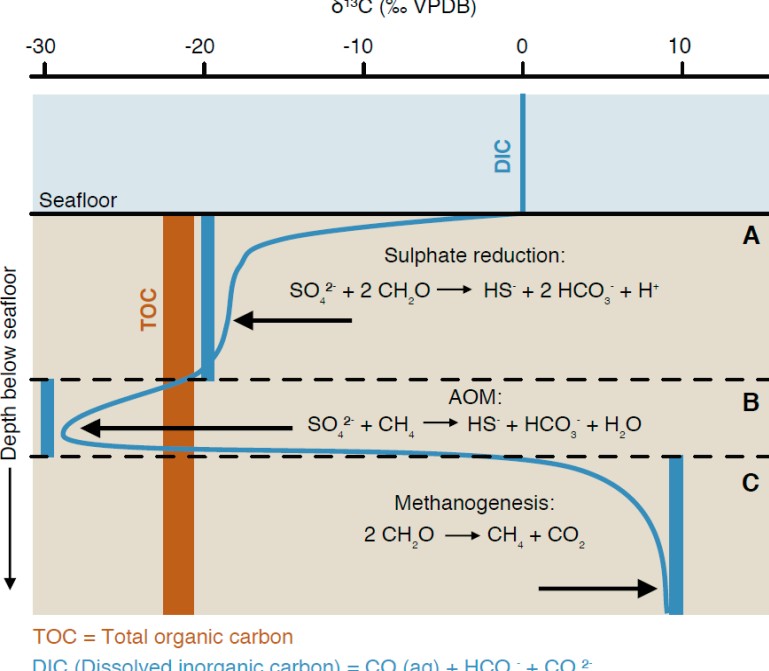

**Figure 4.** Scheme of isotope fractionation effects due to metabolic activity on $\delta^{13}$C of DIC across the SMT. Essentially, three different zones can be distinguished based on metabolic activity and isotopic signatures in the porewater (from top to bottom): (**A**) Sulphate reduction zone; (**B**) SMT showing anaerobic oxidation of methane; (**C**) methanogenic zone. The thick blue bars represent the isotopic compositions of the instantaneously produced metabolic DIC; the thin blue line represents the resulting isotope profile of DIC.

*Isotope profiles at the SMT*: Further complexity arises near the SMT, where even more $^{13}$C-depleted DIC from AOM is mixing with DIC from the sulphate reduction zone (Figure 4B). Because isotopically light methane is quantitatively converted to DIC, $\delta^{13}$C of DIC may become much more negative than $\delta^{13}$C of TOC at the SMTZ. Extremely negative $\delta^{13}C_{DIC}$ values ($<-90$‰) were measured in deep continental crust fracture systems (Drake et al., 2015, 2017 [65,66]). However, such negative values are only possible under DIC limitation, which is not usually the case in marine sediments (see above section). The negative effect from AOM is usually balanced by diffusive mixing with $^{13}$C-enriched DIC from the methanogenic zone (Figure 4C).

*Isotope profiles in the methanogenic zone*: As shown above and in Figure 1A, fractionation effects are rather strong during hydrogenotrophic methanogenesis, resulting in isotopically light $CH_4$ ($\delta^{13}$C $< -60$‰) and isotopically heavy DIC ($\delta^{13}$C $> 0$‰; Figure 4C), whereby some intermediate $CO_2$ is also produced by fermentation (Equation (4)). Acetoclastic methanogenesis may occur, as indicated by enriched $\delta^{13}$C values in acetate observed by Heuer et al. (2008) [7]. However, this process probably does not occur alone and does not explain the large isotopic differences between DIC and $CH_4$ observed in most methanogenic zones. As a result of diffusive mixing, DIC shows a mixing curve between the negative values at the SMT and moderately positive values in the methanogenic zone (Figure 4C).

### 3.2. Towards a Simulation of Carbon-Isotope Profiles

*Closed-system Rayleigh models*: Several studies have been dedicated to understanding these effects using closed-system Rayleigh fractionation models (Nissenbaum et al., 1972 [67]; Claypool and Kaplan, 1974 [6]; Whiticar and Faber, 1986 [19]; and Paull et al., 2000 [68]). $\delta^{13}$C was calculated as a function of the fraction of $CO_2$ converted to $CH_4$ (Rayleigh function), without $CO_2$ production by fermentation, but including the input by $CO_2$ advection and the removal by carbonate precipitation. The model was

solved for separate compartments in a sedimentary porewater profile, yet, the model does not simulate diffusive mixing. It would be incorrect to fit a Rayleigh curve, representing an exponential function, to a porewater profile showing a mixing hyperbola.

*Reaction-diffusion models*: Instead, Alperin et al. (1988) [20] was among the first to use a numerical reaction-transport model to simulate diffusion and fractionation of $CH_4$ above the SMT. By fixing the $\delta^{13}C_{CH4}$ at the SMT and near the sediment surface as boundary conditions, he realized that the shape of the curve in the sulphate-reduction zone is very sensitive to the kinetic fractionation factor of AOM (the forward reaction). Chatterjee et al. (2011) [69] simulated $\delta^{13}C_{DIC}$ through the SMT using a reaction-transport model, which very well reproduced the measured profiles from several sites at Hydrate Ridge and in the Gulf of Mexico.

*Models including isotopic equilibration*: Despite the above studies, it still remained unclear what caused the strongly negative $\delta^{13}C_{CH4}$ values at the SMT. The key insight came from Yoshinaga et al. (2014) [39], demonstrating a reverse flux and isotope equilibration during AOM. These authors could satisfactorily reproduce $\delta^{13}C_{CH4}$ profiles across the SMT at the Cascadia Margin (Figure 5; Bull's eye vent; Pohlman et al., 2008) [70], and this concept was successfully applied to reproduce profiles in the South China Sea (Wu et al., 2018 [71]; Chuang et al., 2019 [72]). Now it is possible to simulate both $\delta^{13}C_{CH4}$ and $\delta^{13}C_{DIC}$ including the complete stoichiometries of metabolic reactions in diffusive sedimentary systems. Meister et al. (2019b) [55] developed and tested such a model system, including the SMT and methanogenic zone over a depth range of 200 m. The modelling confirmed previous insights that isotopic equilibration at the SMT must occur.

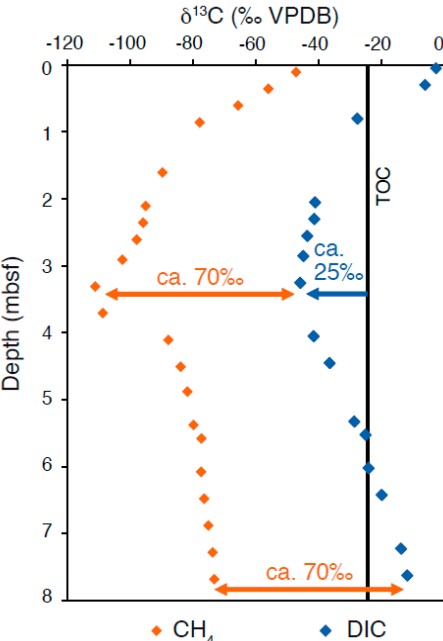

**Figure 5.** Measured $\delta^{13}C_{CH4}$ and $\delta^{13}C_{DIC}$ profiles from SMTs at Site C-2, Bullseye vent, Cascadia Margin (data from Pohlman et al., 2008 [56]; figure modified from Meister et al., 2019b [35]).

*3.3. Assessing the Factors Influencing Carbon-Isotope Profiles*

Having a full reaction-diffusion model at hand, Meister et al. (2019b) [55] discuss the sensitivity of carbon-isotope profiles to variations in organic matter burial and fractionation factors through the SMT and into the methanogenic zone. One observation is that fractionation factors determined in culture experiments with *Methanosarcina barkeri* under different conditions (Krzycki et al., 1987 [26]; Gelwicks et al., 1994 [27]; Londry, et al., 2008 [31]; etc.; Figure 1) result in too small isotopic differences between $CH_4$ and $CO_2$ compared to the differences observed in natural environments. Larger fractionation was observed in cultures using strains of *Methanococcus*, an obligate hydrogenotrophic

methanogen, under substrate limitation (Botz et al., 1996) [28] (Figure 1). These larger fractionation factors would be necessary to reproduce the measured porewater profile (Meister et al., 2019b) [55], including the effect of $CO_2$ production by fermentation as shown by Equation (4).

Overall, methanogenesis results in $^{13}$C-depleted $CH_4$ and $^{13}$C-enriched DIC, and, from a stoichiometric point of view, isotopic values of $CH_4$ and DIC should be symmetrical with respect to the organic matter substrate from which all DIC and $CH_4$ originates, independent of the pathway. The $\delta^{13}C_{CH4}$ and $\delta^{13}C_{DIC}$ approach a symmetrical distribution relative to $\delta^{13}C_{TOC}$, which is indeed observed in many measured isotope profiles (Figure 6A). While the (apparent) separation factor $\varepsilon$ between $\delta^{13}C_{CH4}$ and $\delta^{13}C_{DIC}$ remains constant, the $\delta^{13}C_{CH4}$–$\delta^{13}C_{DIC}$ couple may be shifted to lower values if the organic matter decay is slow or if most organic matter rapidly decays in the sulphate reduction zone (resulting in a deep SMT and curved sulphate profile; Meister et al., 2013a) [73]. This is because the $\delta^{13}C_{DIC}$ would then be more influenced by $^{13}$C-depleted DIC diffusing down from the sulphate-reduction zone (Figure 2B). However, no shift to more positive values (as seen in Figure 6B; Heuer et al., 2009) [7] can ever occur in a steady-state diffusive system. In general, it is difficult to reproduce $\delta^{13}C_{DIC}$ more positive than about 10‰ and impossible to produce the 35‰ observed by Heuer et al. (2009) [7].

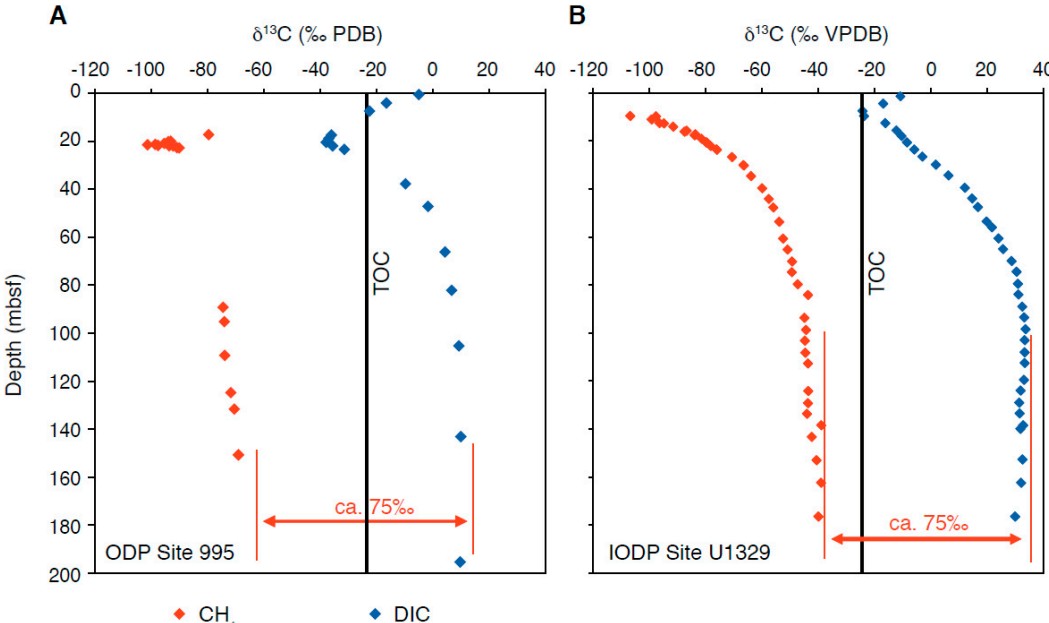

**Figure 6.** Measured $\delta^{13}C_{CH4}$ and $\delta^{13}C_{DIC}$ profiles from natural methanogenic zones: (**A**) Blake Ridge, ODP Site 995 (Paull et al., 2000) [54]: The profiles largely approach symmetry with respect to $\delta^{13}C_{TOC}$ in the methanogenic zone. (**B**) Cascadia Margin, IODP Site U1329 (Heuer et al., 2009) [7]: Profiles are shifted towards more positive values, with DIC reaching +35‰ at 100 mbsf.

In conclusion, most features of $\delta^{13}$C profiles of $CH_4$ and DIC in marine porewaters can be well reproduced with numerical reaction-diffusion models, but to explain very positive values in DIC observed in some methanogenic zones further advective or, perhaps, non-steady state conditions need to be taken into account.

### 3.4. Non-Steady-State Effects and Gas Transport

*Non-steady state effects*: Although it is clear that porewater profiles are often not in a steady state due to changes in sediment deposition and microbial activity (e.g., Dale et al., 2008a; Contreras et al., 2013) [59,74], the effects on isotope profiles are still poorly assessed. It is possible that upward and downward migration of the SMT results in a temporary offset of isotope profiles with respect to

the redox zonation. Also, increase and decrease in overall microbial activity may shift the isotopic signatures (Meister et al., 2019b) [55], as will be further discussed below.

*Methane and CO₂ rise*: In organic carbon-rich sediments both $CH_4$- and $CO_2$-gas fugacities may exceed solubility in porewater, which then leads to exsolution of a gas phase. Gas systematics are rather complex, as many factors play a role, while gas transport is often unpredictable, abrupt, and anisotropic, i.e., it follows discrete conduits. Starting from oversaturated $CH_4$, rates of gas bubble formation can be calculated according to Mogollón et al. (2009) [75]. While methane gas saturation is soon reached, $CO_2$ may not readily be oversaturated due to the buffering effect of porewater alkalinity (see discussion below). Nevertheless, once a gas phase is present, as methane bubbles, $CO_2$ readily equilibrates with the gas phase, following Le Chatelier's principle (Smrzka et al., 2015) [76]. While parts of the methane bubbles adhere to the sediment particles by capillary forces, the rise velocity of methane bubbles can be calculated using the formulations of Boudreau (2012) [77]. Thereby, it is critical to consider that methane bubble rise is facilitated if conduits, such as fractures or porous sediment, are present. Once the bubbles reach a zone with a lower $CH_4$ fugacity near the SMT, bubbles start to re-dissolve into the aqueous phase, but depending on the dissolution kinetics (see Mogollón et al., 2009) [75] and the thickness of the sulphate zone, parts of the methane bubbles may reach the seafloor and escape to the water column (seepage; e.g., Dale et al., 2008b) [78]. While methane seepage out of marine sediments is not commonly reported to be $CO_2$-rich, parts of the $CO_2$ are probably re-dissolved due to the pH-buffering effect of AOM at the SMT. The diffusion constant of $CO_2$ only shows minor dependence on isotopic mass (Zeebe et al., 2011) [79]; however, the fact that $CO_2$, which is depleted in $^{13}$C by ca. 9‰ relative to $HCO_3^-$, rises more rapidly (due to a higher diffusion constant, or in the gas phase) may also have an effect on the overall isotopic composition of DIC. However, this effect is limited by the carbonate equilibrium, preventing a runaway Rayleigh effect.

*Deep methane sources*: Methane gas may not only evolve within the considered sediment sequence, but may rise from deeper intervals of organic carbon-rich sediments, from the gas hydrate stability zone (e.g., Borowski et al., 1999; Kennett et al., 2000) [80,81], or even from the thermogenic zone, where methane is generated from the thermogenic decomposition of organic matter. In particular, thermogenic methane is generally less depleted in $^{13}$C (e.g., Whiticar, 1999) [82]. It still remains unexplained how DIC with extremely positive $\delta^{13}C_{DIC}$ values could be produced in the methanogenic zone. However, if thermogenic methane could re-equilibrate, isotopically, with the DIC pool through the WL-pathway, this could explain the positive values observed at the Cascadia margin (Heuer et al., 2009) [7], where the $CH_4$-DIC couple is shifted to more positive $\delta^{13}$C than symmetry with respect to TOC (Figure 6B). Testing such scenarios requires more elaborate numerical models under non-steady-state conditions.

## 4. The Isotope Signature of Diagenetic Carbonates

### 4.1. Processes Inducing Carbonate Formation in the Deep Biosphere

Before discussing how carbon-isotope patterns characteristic for particular biogeochemical zones are preserved in diagenetic carbonates, the factors controlling authigenic (incl. diagenetic) carbonate formation in sediments need to be briefly summarized. Generally, carbonates precipitate due to an increase of the saturation state (here expressed as saturation index $SI = \log IAP - \log K_{SP}$, where IAP is the ion activity product and $K_{SP}$ is the solubility product). Most commonly, cations (mainly $Ca^{2+}$ and $Mg^{2+}$) are sufficiently supplied from seawater in the uppermost few metres (Baker and Burns, 1985; Meister et al., 2007) [11,83] or sometimes through deep circulating fluids (Meister et al., 2011) [84]. Carbonate saturation can be significantly increased due to microbial metabolic activity, whereby it is still debated which microbial processes indeed can induce carbonate precipitation. Under marine conditions, sulphate reduction, which is producing DIC and alkalinity at a 1:1 ratio, may even lead to a lowering of the saturation state due to a drop in pH (Meister, 2013 [85]; and references therein), unless most of the 28 mmol/L of sulphate in seawater are turned over. In contrast, AOM produces two moles of alkalinity per mole of DIC and, thus, efficiently increases the SI of carbonates (Moore et al.,

2004; Ussler III. et al., 2008; Meister, 2013) [85–87]. Furthermore, methanogenesis always produces $CO_2$ and no alkalinity (as seen from Equation (3)), but the acidification effect is largely buffered by alkalinity produced near the SMT and the release of ammonia. Further alkalinity may originate from the alteration of silicates (mainly volcanic glass but possibly also by clay minerals; Wallmann et al., 2008; Meister et al., 2011; Wehrmann et al., 2016) [84,88,89]. These sources of alkalinity production, perhaps in combination with exsolution of $CO_2$ via methane bubbles, may prevent carbonate undersaturation in the methanogenic zone. Although these effects have not been precisely quantified yet, and it remains unclear how focused diagenetic beds of carbonate can form in the methanogenic zone, it is most likely due to the dynamics of a supersaturation front (cf. Moore et al., 2004) [86].

### 4.2. Controls of $\delta^{13}C$ Composition of Diagenetic Carbonates

Carbonate precipitation itself is subject to fractionation effects, whereby equilibrium fractionation prevails at slow precipitation rates observed in the deep biosphere (Turner, 1982) [90]. The carbonate mineral phase is, in most cases, enriched in $^{13}C$ by a few permil relative to the inorganic carbon (~2‰ for calcite; Deines et al., 1974) [91]. For dolomite, the separation factor relative to $CO_2$ is on the order of 12‰–14‰ at ambient temperatures (Ohmoto and Rye, 1979; Golyshev et al., 1981) [92,93]. Subtracting the isotope effect of 9‰ between $CO_2$ and $HCO_3^-$ (Mook, 1974) [94] results in a range of 3‰–5‰ for dolomite and $HCO_3^-$. As a result of this fractionation effect, it was suggested that the residual DIC is depleted in $^{13}C$ due to a Rayleigh effect (Michaelis et al., 1985) [95]. However, since carbonate precipitation is limited by the production of alkalinity and the supply of major cations from seawater, while DIC is usually not limited in the deep biosphere, as it is produced in ample amounts from microbial dissimilation reactions, carbonate precipitation has most likely a minor effect on the isotopic composition of DIC. This has also been confirmed by model calculations (e.g., Chuang et al., 2019) [72]. Thus, the carbon-isotope signature of the porewater becomes trapped in the diagenetic carbonate, providing a signature for past biogeochemical conditions at the location and time of precipitation.

*Suboxic vs. anoxic zones*: It is often seen in carbonates, especially if they occur in organic carbon-rich sediments, that $\delta^{13}C$ values are in a range between 0‰ and −10‰, but not as negative as to indicate a signature typical for a sulphate reduction zone. This could be the result of a precipitation in the top few centimetres below the sediment surface, where carbon isotopes follow a mixing hyperbola. Typically, in suboxic sediments, where the dissimilatory rates are moderate, and the redox zonation accordingly expanded, $\delta^{13}C$ values fall into this intermediate range, as observed in Ca-rich rhodochrosite occurring within mottled and bioturbated sediments of the Eastern Equatorial Pacific (Meister et al., 2009) [96]. Thereby Fe- and Mn-reduction may contribute to carbonate supersaturation (Kasina et al., 2017, and references therein) [97]. Intermediate values may also occur in shallow sulphate-reduction zones, e.g., in bituminous sediments, where laminae of authigenic carbonate form just below the sediment/water interface and, hence, early with respect to burial along the mixing gradient (cf. the Triassic Besano Fm., Ticino, Switzerland; Bernasconi et al., 1994 [98]; see discussion in Meister et al., 2013b) [99]. Alternatively, the isotope values may represent a mixture of different carbonate phases of different origin, e.g., dolomite mud from an adjacent platform, showing normal marine isotope values. Therefore, a further petrographic analysis is often necessary to determine the origin of the carbonate, in order to interpret its carbon-isotope signature.

*Sulphate-methane transition zone*: While sulphate reduction alone rather lowers the saturation state of carbonates, an early onset of AOM has been suggested to induce the formation of carbonates, such as carbonate concretions in organic carbon-rich shale of the Santana Fm. (Brazil; Heimhofer et al., 2018) [100] at very shallow depths. Shallow SMT zones are well documented from modern settings (e.g., Thang et al., 2013) [101] and they may indeed induce carbonate cementation (e.g., Jørgensen et al., 1992) [102]. However, an actual AOM signature in $\delta^{13}C$ with values below −35‰ is only exceptionally preserved, such as in a dolomite layer at the Peru Trench at 6.5 m below seafloor (mbsf; Meister et al., 2007) [11]. Instead, a great range of carbon-isotope values have been reported from dolomite layers intercalated in organic carbon-rich diatom ooze drilled from upwelling regions offshore California

(Pisciotto and Mahoney, 1981) [103], in the Gulf of California (Kelts and McKenzie, 1982) [104], or in Miocene diatomite of the Monterey Fm. (California; Murata et al., 1969; Kelts and McKenzie; 1984) [9,105]. Also, Rodriguez et al. (2000) [106] report strongly positive $\delta^{13}C$ values in siderites from a methanogenic zone at Blake Ridge. While the positive $\delta^{13}C$ values were interpreted as a result of precipitation in the methanogenic zone, negative values in dolomites of the Messinan Tripoli Fm. in Sicily were interpreted as indicative of precipitation in the sulphate-reduction zone. This explanation seems obvious, but it still remains unclear what caused precipitation of carbonates in the methanogenic zone, as methanogenesis should not per se lead to a focused supersaturation of carbonates.

*Deep methanogenic zone*: A case in which dolomite cements are indeed observed to form in the methanogenic zone is OPD Site 1230, located in the Peru Trench. This site, at a water depth of 5000 m, is located on the Peruvian accretionary prism, where the sedimentary succession is dissected by a fault zone at 230 mbsf). A dolomite breccia was drilled at this depth, showing more radiogenic $^{87}Sr/^{86}Sr$ ratios than modern seawater, indicating precipitation from a fluid that was derived from interaction with continental basement rocks, deep in the prism (Meister et al., 2011) [84], presumably delivering alkalinity and $Ca^{2+}$ to induce dolomite precipitation. While this site represents a special case, it is still not understandable how dolomite can otherwise form in a methanogenic zone.

## 4.3. Interpreting $\delta^{13}C$ Archives Through Time

Carbon isotopes were measured in diagenetic dolomites through a 150 m thick interval on the Peru Margin, showing varying $\delta^{13}C$ (Figure 7A–C; ODP Site 1229; Meister et al., 2007) [11]. The dolomites also show $^{87}Sr/^{86}Sr$ ratios near to Pleistocene–Holocene seawater, while the ratios in the porewater strongly decrease with depth, indicating that the dolomites formed near to the sediment– water interface. The dolomites thus formed in the past and document an actively evolving biosphere through time.

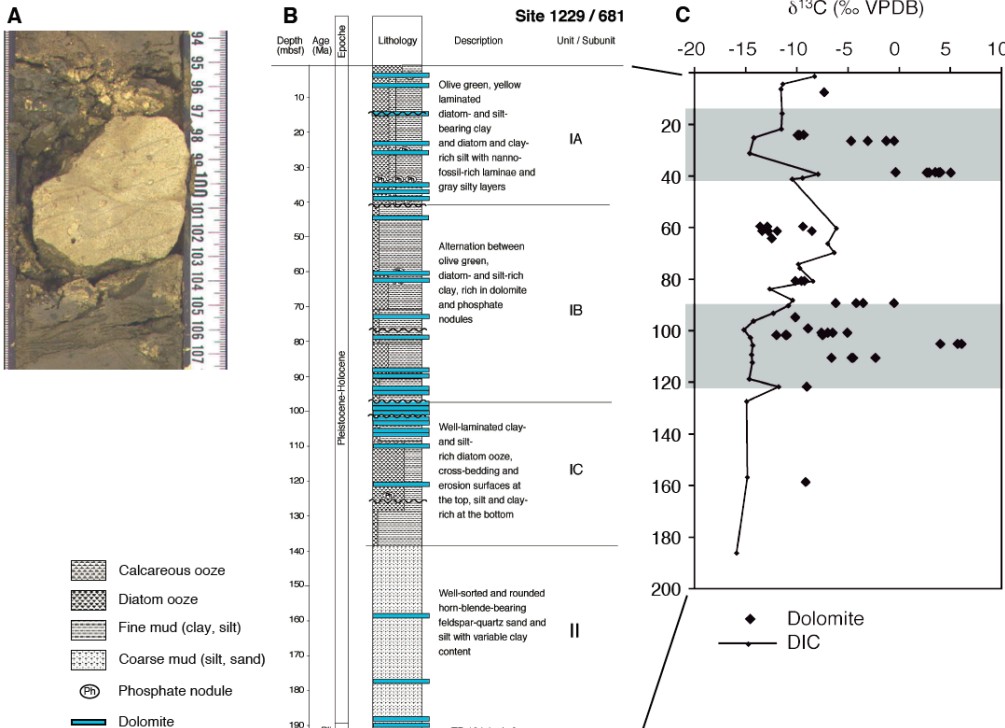

**Figure 7.** Patterns of carbon isotopes preserved in diagenetic carbonates from the Peru Margin: (**A**) Fragment of discrete, hard lithified, diagenetic dolomite; (**B**) distribution of dolomite layers through the sequence at ODP Site 1229; (**C**) carbon-isotope values in diagenetic dolomites in comparison to carbon-isotope composition of the present porewater.

*Episodic carbonate formation*: The dolomite beds do not present uniform conditions through time, as otherwise fine-grained dolomite would have been homogeneously distributed throughout the sediment. Instead precipitation must have occurred episodically, at focused locations. Based on their regular spacing on the order of glacial–interglacial cycles in the sediment, Compton (1988) [107] proposed that diagenetic dolomite beds in the Monterey Fm. could be linked to Milankovitch cyclicity in sediment deposition. Meister et al. (2008) [108] followed this idea by showing that oxygen isotopes in dolomites, that occur with a spacing of ca. 10 m, reflect marine $\delta^{18}O$ values and bottom water temperatures of glacial periods on the Peru Margin. Contreras et al. (2013) [59] found enrichments of dolomite, barite, and isotopically light archaeol ($\delta^{13}C = -73‰$) as an imprint of an earlier shallow SMT zone far above the present SMT, which has shifted downward since then. In this manner, the isotopic signature of archaeol can be explained by fractionation as part of the Wood–Ljungdahl pathway (see discussion above). Reaction-transport modelling then confirmed that an upward and downward migration over 30 meters within the time frame of 100 ka is feasible. In conclusion, the dolomite layers on the Peru Margin formed in the aftermath of a rapid deposition of an organic carbon-rich, interglacial sediment layer that, during its burial, triggered a temporary onset of a shallow SMT zone. During this time, a dolomite layer formed at the upper SMT.

*Dynamics in carbon-isotope preservation*: While the episodic precipitation of dolomites can be explained by a 100 ka cyclicity in deep biosphere activity, longer-term changes on the order of several 100 ka are superimposed and manifested in the $\delta^{13}C$ record. In theory, three explanations can be proposed (Figure 8A–C): (A) Precipitation in different zones as suggested by Kelts and McKenzie (1984) [9], whereby the carbon-isotope signature is controlled by the depth of a carbonate saturation front relative to the redox zonation and $\delta^{13}C$ profile in the porewater. The saturation front could be uncoupled from the redox zonation due to outgassing of $CO_2$ from the methanogenic zone. (B) At a shallow SMT, outgassing of $CH_4$ is frequently observed (e.g., Dale et al., 2008b) [78], resulting in the loss of $^{13}C$-depleted carbon and accordingly, less negative $\delta^{13}C_{DIC}$ near the SMT. This mechanism can be well reproduced with reaction transport modelling (Meister et al., 2019b, Figure 7 therein) [55]. (C) Due to an increase in methanogenic activity, $\delta^{13}C$ in both $CH_4$ and DIC may increase in the methanogenic zone. This is clearly the case at Peru Margin ODP Site 1229, where the modern-day $\delta^{13}C_{DIC}$ is negative in the methanogenic zone, but was positive in the past (Meister et al., 2019a; and references therein) [13]. The observation that the variations of $\delta^{13}C$ in diagenetic dolomites are coupled to variations of $\delta^{34}S$ in pyrite provides independent evidence that two episodes of enhanced deep-biosphere activity involving stronger methanogenic activity occurred throughout the Pleistocene. Most likely, a combination of the effects A-C occurs in a dynamic way. Ultimately, reaction-transport modelling under non-steady-state conditions will clarify the mechanisms that generated these diagenetic records.

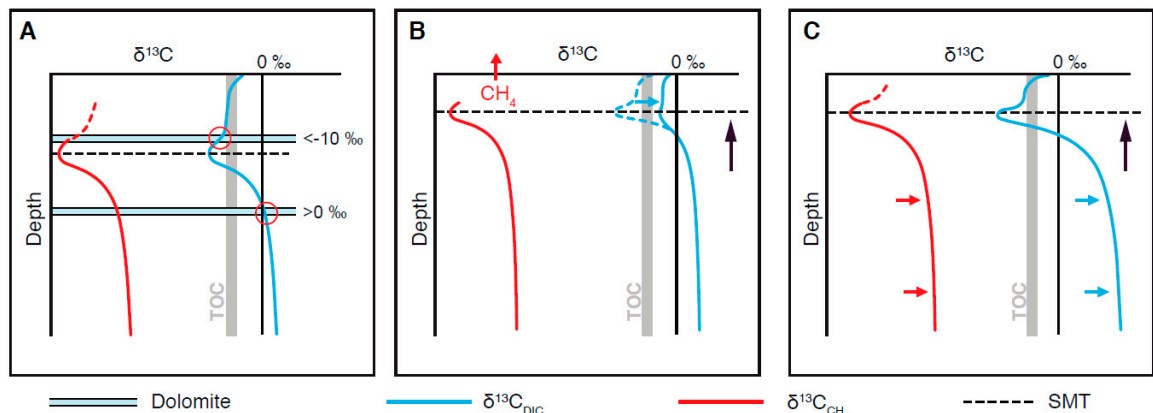

**Figure 8.** Possible scenarios of carbon-isotope incorporation in diagenetic dolomites: (**A**) Dolomite precipitation in different redox zones; (**B**) dolomite formation at the SMT showing variable $\delta^{13}C_{DIC}$ as a result of $CH_4$ escape; (**C**) changing $\delta^{13}C_{DIC}$ due to variations in methanogenic activity.

## 5. Implications for the Global Carbon Cycle

*5.1. The Importance of Diagenetic Carbonates for Global Carbon Fluxes*

Carbonates are besides sedimentary organic carbon the largest sink of $CO_2$ from the exogenic cycle. The capacity of the ocean to precipitate carbonate relies on the delivery of alkalinity from continental silicate weathering. Also, a significant portion of carbonate forms as a result of submarine alteration of ultramafic rocks, so that subduction of ophicarbonates substantially contributes to the global carbon cycle (e.g., Alt and Teagle, 1999) [109]. Diagenetic carbonates forming as a result of dissimilatory microbial activity in marine sediments provide a further carbon sink. Their formation does not rely on alkalinity supplied by continental silicate weathering; they are induced by anaerobic terminal electron accepting processes acting as an alkalinity pump. This alkalinity remains in the sediment, as long as sulphide is trapped as iron sulphides and not re-oxidized at the sediment surface. Microbial alkalinity production contributes to retaining a significant portion of DIC, derived from organic matter, that otherwise would be cycled back to the water column and atmosphere (Figure 9). Schrag et al. (2013) [8] estimated that diagenetic carbonates significantly contribute to the global carbonate burial flux, in particular at times of widespread anoxia, and that this flux also may substantially affect global carbon-isotope composition. Burial of isotopically light carbon, as diagenetic carbonate, would contribute to an increase in $\delta^{13}C$ in ocean and atmosphere. In turn, formation of isotopically heavy diagenetic carbonates would decrease $\delta^{13}C$ in the ocean and atmosphere, and therefore, understanding the controls of methanogenic carbonate formation would be significant for assessing the deep biosphere's influence on the carbon cycle.

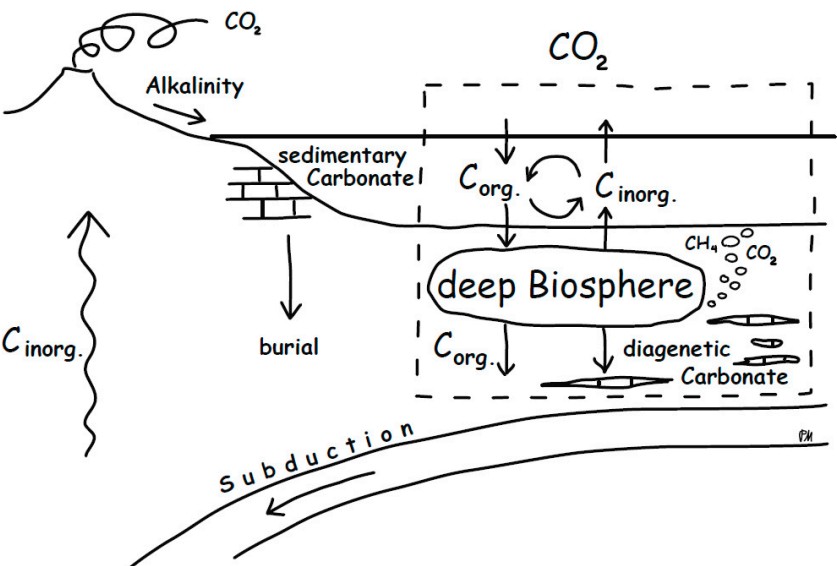

**Figure 9.** Schematic drawing representing the role of the deep biosphere in the global carbon cycle. Parts of the organic carbon deposited on the seafloor are re-mineralized and cycled back to the water column. Some inorganic carbon is buried as diagenetic carbonate. The latter is mainly induced by additional alkalinity from anaerobic metabolisms and is decoupled from sedimentary carbonate induced by alkalinity from continental weathering. Most carbon is stored in the rock record, but the deep biosphere has a significant effect on what goes into this reservoir.

*5.2. Diagenetic vs. Atmospheric Signatures?*

Certain time periods in Earth's history stand out due to significant excursions in the global carbon-isotope composition. These excursions may be largely influenced by the activity of the deep biosphere. For example, methane released from the dissociation of gas hydrates as a result of warming of bottom water during the Palaeocene–Eocene thermal maximum has caused a perturbation in the carbon

cycle, manifested in a negative excursion in the atmospheric $\delta^{13}C$ (Dickens, 1997) [110]. Also, positive excursions occurred during the Neoproterozoic (Knoll et al., 1986) [111] or near the Permian–Triassic boundary (Payne et al., 2004) [112], presumably due to enhanced burial of organic carbon.

While signatures in the carbon-isotope record can often be globally correlated, it still has to be confirmed that these are not results of diagenetic overprint. One indication that the signatures indeed represent the conditions in the water column would be that isotope records in the organic carbon show a similar excursion as the inorganic carbon, i.e., the organic and inorganic fractions run in parallel, offset by 20‰–30‰, depending on the separation factor of the primary production (Knoll et al., 1986) [111]. Isotopically light signatures in organic carbon from the Peru Margin were initially interpreted as signatures of a large-scale methane escape to the water column along the Peru Margin, presumably as a result of gas hydrate dissociation (Wefer et al., 1994) [113]. However, the finding of accumulated isotopically light archaeol, in combination with barite and dolomite enrichments, supports the concept that these signatures instead represent the imprint of a former SMT (Contreras et al., 2013) [59] and are thus a diagenetic feature. Similarly, Louis-Schmid et al. (2007) [114] found that a 4‰ negative excursion in a Late Jurassic hemipelagic succession at Beauvoisin (SE France) is the result of local anaerobic methane oxidation and precipitation of authigenic carbonates, rather than a global signal.

An even more extreme carbon-isotope excursion can be found in the Palaeoproterozoic record. The so-called Lomagundi–Jatuli event (ca. 2.2–2.1 Ga) lasted more than 100 Ma and is the most positive excursion in $\delta^{13}C$ (up to +14‰) in Earth's history. It was thought that this excursion is the result of strongly enhanced organic carbon burial rates as a result of the onset of oxygenic photosynthesis at the time of the great oxidation event, i.e., when Earth's atmosphere first became oxidized (Schidlowski et al., 1984) [115]. However, the excursion is not observed in the organic carbon record, which casts some doubt on the explanation that organic carbon burial is responsible for such a large isotope excursion. Hayes and Waldbauer (2006) [116] challenged this interpretation, using a global model of carbon and energy fluxes. According to their calculation, such a large carbon burial would have resulted in unrealistically high oxygen concentrations (up to 100 times the present level). Instead, they suggested that organic carbon production was enhanced and, while the sulphur cycle was not fully developed yet, organic matter would have been converted to $CH_4$ and $CO_2$ at shallow depths, with minimal AOM taking place. Escape of isotopically light methane would have led to an isotopically heavier residual DIC pool. Presumably, $CO_2$ outgassing was partially prevented by a high alkalinity of the ocean due to silicate weathering, causing carbonates to precipitate. In other words, the signature observed in the carbonate record represents a diagenetic signal and, since this signal occurs simultaneously around the Earth, it would represent a "global diagenetic event".

That this process indeed can occur has been demonstrated for modern stromatolites in a brackish to seasonally evaporative, coastal pond in Brazil (Lagoa Salgada), showing extremely positive carbon-isotope signatures of >15‰ (Birgel et al., 2015) [117]. In fact, this system has been suggested as a modern analogue to explain the large excursions in $\delta^{13}C$ in the aftermath of the GOE. Also, in this case, carbon-isotope signatures provide an archive of major dynamics in the "shallow" deep biosphere of global scale.

## 6. Conclusions

This synopsis provides an overview of processes affecting carbon isotopes preserved in diagenetic carbonates and the potential of these isotopic signatures to serve as indicators of past conditions in the marine deep biosphere. Culture experiments using different organisms under a range of different growth conditions, as well as radiocarbon tracer experiments, provide insight on potential fractionation mechanisms as part of the Wood–Ljungdahl pathway, which is the main methane metabolizing pathway. It is indicated that during AOM, but also during methanogenesis, isotopic equilibration between $CO_2$ and $CH_4$ could play a role and could explain natural isotope distributions, which are near to a distribution expected at thermodynamic equilibrium. Taking these effects into account, isotopic

distributions in diffusive porewater profiles, in response to organic matter content, reactivity, and sedimentation rate, are largely reproducible using numerical reaction-transport models.

Carbon-isotope signatures preserved in diagenetic carbonates show strongly variable values, but they are still not fully understood. Most likely, diagenetic carbonates form episodically, reflecting dynamic conditions and probably upward and downward shifts in the redox zonation. Most likely, activity of the deep biosphere is linked to cyclicity in oceanographic conditions and sediment deposition on different timescales, partially affecting the isotopic signatures. Advective transport of gas phase $CH_4$ and $CO_2$ may also affect the carbonate isotope profiles. Carbonate precipitation is generally induced by alkalinity production by microbial activity, most importantly, AOM and ammonium release, but perhaps also silicate alteration. Upon discharge of $CO_2$ from the methanogenic zone, the supersaturation front may be decoupled from the depth of the SMT. Most urgently, though, numerical models under non-steady-state conditions are needed to disentangle such dynamics and to resolve peculiar carbon-isotope distributions in the deep-time geological record. This will ultimately reveal the potential of carbon-isotope signatures in diagenetic carbonates as archives of past deep-biosphere conditions.

**Author Contributions:** P.M. contributed the geochemical parts. C.R. contributed the biochemistry parts.

**Funding:** The work was partially supported by the through the Marie Skłodowska-Curie Actions Research Fellowship Programme: Project TRIADOL (grant agreement no. 626025) and the Department of Geodynamics and Sedimentology at the University of Vienna. Open Access Funding by the University of Vienna.

**Acknowledgments:** We thank S. Rittmann for comments related to the Wood-Ljungdahl pathway, as well as M.E. Böttcher, B. Liu, and B.B. Jørgensen for insightful discussions. We also thank three anonymous reviewers for constructive comments to the manuscript.

**Conflicts of Interest:** The authors declare no conflict of interest.

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
