# Peer review of "The Carbon-Isotope Record of the Sub-Seafloor Biosphere"

_geosciences, doi:10.3390/geosciences9120507_

Round 1

Reviewer 1 Report

Thanks for the revised version of the manuscript. The subheadings and clarifications greatly improve the flow.

Reviewer 2 Report

Revised well from the original version.

This manuscript is a resubmission of an earlier submission. The following is a list of the peer review reports and author responses from that submission.

Round 1

Reviewer 1 Report

Meister and Reyes have presented a timely review relating to carbon isotopes in the deep biosphere and how they may be preserved in carbonates.
I think the topic is relevant and the review needed, however it needs to be re-worked to be understandable for a wider readership. It is currently very targeted towards isotope chemists, but the second half of the paper links to carbonate precipitation and the microbial influence on this, which is a hot topic in geomicrobiology and will attract a general geomicrobiology audience as well, so needs to be understandable to a general geomicrobiologist.

I found the flow very difficult, in almost all sections. It seemed like the authors were jumping from study to study and the main point of each section wasn't always clear in amongst the abundance of detail and listed studies.
I recommend including subsection headings to further break down the existing sections in order to make the flow logical. And/or including a key sentence at the start of each section or each paragraph, stating the main point of that section. After that sentence, subsequent examples from the literature will then be supporting evidence for that statement/idea, and this will make the review easier to follow. At present the reader has to retain a lot of information in the forefront of their mind in each section, following study after study or point by point, and not being quite sure what the key takehome message is in each section.

Specific comments by section:
line 96: I would replace (1) with 'firstly' since the numbers of equations here make this confusing. and since the 'second' approach is quite some distance away in text...

I found section 2.1 confusing (starting around line 98) as the introduction/abstract seemed to suggest that the focus would be on fractionation relating to methane cycling (and acetogenesis). However, in section 2.1, the dissimilatory sulphate reduction is the first reaction microbial process to. Also, in the 'second approach' of section 2.1, the jumping between sulphate reduction and methane cycling makes this confusing. Perhaps a sentence or two, or even a background section on the main microbial processes in natural porewater before section 2.1, might help make this section flow better and might make the jumps between sulphate reduction and methane cycling more easy to follow.

For a broad readership including people interested in geomicrobial processes, but not necessarily isotope geochemists, I think the interchange between 'separation factor' and 'fractionation factor' will be confusing (line 111). e.g. I don't think most geomicrobiologists would know what 'separation factor' is without some context, so either give it better context before using it line 111 or explain it somewhere like: the extent of isotopic fractionation can be expressed by a fractionation factor, epsilon, also known as a separation factor... or similar

Fig 1A is unclear at present (what is the y axis? abundance of what?)

line 176: check spellnig of Methanococcus
line 179: spelling of suggest

section 2.2 - the section starts with AOM as the lead into equilibrium fractionation. It seems unusual to introduce methanogenesis at line 173-180 and then jump back to an example about AOM (lines 181-188 and Fig 2). Reorder this so that methanogenesis isn't in between two sections of text about AOM. Or spell it out in text that Fig 2A is an example using AOM and also clarify in the text that Wegener is an example from AOM.

Line 206 and Fig 3: I recommend to use the enzyme name 'Formate dehydrogenase' rather than 'molybdenum complex' (line 206) as MoCo which is referred to as molybdenum pyranopterin in the figure caption (Fig 3) is not an enzyme, it is a cofactor.

Figure 3: please have a separate diagram (Fig 3D) for bacterial Wood Ljungdahl as it is unnecessarily complicated having to look for it amongs the methane metabolisms. Also, what is the blue line in Fig 3A? and what is the dashed line in Fig3B? These are not indicated in the caption and need to be (they are later described in the main text).

section 3.1 needs to be broken up into headings to make the flow clearer. It was lot of text and the flow was not always easy to follow
Actually, I recommend subsection headings throughout all of section 3 and 4 as there is a lot of detail to have to retain in the short term memory, thus it is very easy to get lost, and difficult to see/remember what the main point of each current subsection is. If the authors can break these subsections down into succinct subheadings, that will help the reader follow the flow and focus on one idea at a time, rather than having to retain the details presented.

Conclusions
I recommend to re-word 'new insights from culture experiments...' as it suggests that new data are reported in this paper (but I don't believe that is the case, just a reference to recent published radiotracer experiments)

Author Response

Please find the fully layouted point-by-point response to reviewers including a cover letter attached.

Reviewer 2 Report

The manuscript (ID: geosciences-609929) reviews carbon isotope systematics in marine sediment environment. In general, both biogeochemical mechanisms and natural distributions are covered and interpreted in detail. As the manuscript is helpful for scientists in early career and other study fields to understand carbon (isotope )behaviors in sedimentary biogeochemistry, I recommend the manuscript to be published on Geosciences after minor revision. Followings are suggestion for revision.

I can find some jargons for isotope geochemists, such as light isotope fractionation, for example. For this issue, “An introduction to isotopic calculations” by John Hayes or some others are helpful.  

Several important papers about microbiological studies that investigate isotope effects on methanogenesis and reversibility of methanogenesis are absent. I guess that Valentine et al. 2004 GCA, Penning et al. 2005 GBC, Moran et al. 2005 Archea, Takai et al. 2008 PNAS, and Okumura et al. 2016 Prog Earth Planet Sci should be referred. For aceticlastic methanogenesis, Sugimoto & Wada 1995 GCA is a milestone for connecting intramolecular isotope heterogeneity in organic matter and isotope fractionation in metabolic products should be cited.

Abundance and isotope profiles of carbon species in sedimentary deep biosphere have been reported frequently. As I know, Ijiri et al. 2012 Org. Geochem. and Ijiri et al. 2013 Res. Org. Geochem. can be referred. 

Figure 9 is nice. I like it.

Reviewer 3 Report

Dear Authors,

The manuscript “The carbon isotope record of the deep biosphere” comes with a plethora of references and offers a great overview on the current understanding of carbon isotope fractionation as well as on open questions and challenges for the analysis of carbon signatures. I really liked the discussion on the isotope signatures of diagenetic carbonates, in particular, section 3.2 and 4.

The manuscript is well-written, but it is a bit dense for those, like me, not fully familiar with the topic and literature. One aspect that I struggled with was to find all the links in the different sections to the major biochemical pathway being considered to discuss fractionation – the WL pathway. Despite the extensive review and great discussions presented, I felt that the focus was often loss and connections between WL and fractionation were not clear. I admit that the main reason for this feeling could be my lack of understanding of all the concepts being discussed in this paper, but it would be great if improvements could be made to make it clearer and more accessible to a wider research community.

In any case, I am positive that this is a great contribution to the field, and I am in favor of its publication.

Specific comments:

Line 199 -  consider replacing by “While radiotracer experiments (…)” to avoid word repetition.

Line 512 – Remove (see discussion below)

Line 620 – Font size

Figure 9 – Nice drawing, but better if edited with vector graphic software.
